# Stakeholder Views of the Proposed Introduction of Next Generation Sequencing into the Cystic Fibrosis Screening Protocol in England

**DOI:** 10.3390/ijns10010013

**Published:** 2024-02-14

**Authors:** Pru Holder, Corinna C. Clark, Louise Moody, Felicity K. Boardman, Jacqui Cowlard, Lorna Allen, Claire Walter, James R. Bonham, Jane Chudleigh

**Affiliations:** 1Florence Nightingale Faculty of Nursing, Midwifery & Palliative Care, King’s College London, London SE5 9PJ, UK; jane.2.chudleigh@kcl.ac.uk; 2Warwick Medical School, Warwick University, Coventry CV4 7AL, UK; corinna.clark@warwick.ac.uk (C.C.C.); felicity.boardman@warwick.ac.uk (F.K.B.); 3Centre for Arts, Memory and Communities, Coventry University, Coventry CV1 5FB, UK; l.moody@coventry.ac.uk; 4Paediatric Respiratory Medicine, Royal London Children’s Hospital, London E1 1FR, UK; j.cowlard@nhs.net; 5Cystic Fibrosis Trust, London EC3N 1RE, UK; lorna.allen@cysticfibrosis.org.uk (L.A.); claire.walter@cysticfibrosis.org.uk (C.W.); 6Pharmacy, Diagnostics and Genetics, Sheffield Children’s NHS Foundation Trust, Sheffield S10 2TH, UK; j.bonham@nhs.net

**Keywords:** cystic fibrosis, next generation sequencing, genomics, CRMS/CFSPID

## Abstract

The project aimed to gather, analyse, and compare the views of stakeholders about the proposed UK cystic fibrosis (CF) screening protocol incorporating next generation sequencing (NGS). The study design was based on principles of Q-methodology with a willingness-to-pay exercise. Participants were recruited from 12 CF centres in the UK. The study contained twenty-eight adults who have experience with CF (parents of children with CF (*n* = 21), including parents of children with CF transmembrane conductance regulator (CFTR)-related metabolic syndrome (CRMS)/CF screen positive—inconclusive diagnosis (CFSPID), an uncertain outcome (*n* = 3), and adults with CF (*n* = 4)), and nine health professionals involved in caring for children with CF. Parents and health professionals expressed a preference for a sensitive approach to NGS. This was influenced by the importance participants placed on not missing any children with CF via screening and the balance of harm between missing a case of CF compared to picking up more children with an uncertain outcome (CRMS/CFSPID). Given the preference for a sensitive approach, the need for adequate explanations about potential outcomes including uncertainty (CFSPID) at the time of screening was emphasized. More research is needed to inform definitive guidelines for managing children with an uncertain outcome following CF screening.

## 1. Introduction

Newborn bloodspot screening (NBS) for cystic fibrosis (CF) became part of the national programme in the UK in 2007. The algorithm uses immunoreactive trypsin (IRT) as a first-line biochemical test, followed by DNA analysis. The first DNA panel consists of four common cystic fibrosis transmembrane conductance regulator (CFTR) mutations. If two of these ‘disease-causing mutations’ are identified, they are referred to as ‘CF suspected’. If only one mutation is identified, genetic testing of a larger panel of 50–100 mutations is undertaken. The existing algorithm also includes a “safety net” for infants with a high IRT but no recognised mutation, who are then referred for further testing [1]. Following NBS, from the annual total of approximately 650k babies tested in England, approximately 150 babies are referred to clinical teams as ‘CF suspected’. A further 250 babies require a second sample to be obtained. For 200 of these, the second IRT is normal, and the families are notified that the baby is likely to be a carrier. In the remaining 50, the IRT remains elevated, and the baby is referred as ‘CF suspected’. A further 50 babies also enter the ‘safety net’ arm of the algorithm. A total of 10 of these turn out to have a lower IRT on repeat, and are reported as CF ‘not suspected’, and 40* have an elevated IRT on repeat, and are referred as ‘CF suspected’ [1]. When a baby is reported as ‘CF suspected’, a clinical referral is made to undertake confirmatory testing. This may result in one of three possible re-classifications:(i)‘CF not suspected’: a false positive screening test result; the baby is then discharged.(ii)‘CF confirmed’: a true positive; appropriate on-going treatment is then arranged.(iii)‘CF transmembrane conductance regulator (CFTR)-related metabolic syndrome (CRMS)/CF screen positive—inconclusive diagnosis’ (CFSPID): the result is positive, but the confirmatory tests are equivocal and there is insufficient information to classify the baby as having CF, but neither is there sufficient clarity to discharge the baby. In these situations, some on-going monitoring is usually arranged [1,2].

This algorithm works well but has some disadvantages. These include the following: (1) the reporting of carriers (around 200 per annum); (2) the need for repeat samples (around 300 per annum), which can be distressing for parents and is also quite costly to the service; (3) some babies are designated as CRMS/CFSPID, and this results in ongoing uncertainty for the parents and the child as they grow up. To address the disadvantages associated with the existing algorithm, replacing the limited panel of mutations (following an elevated IRT) with an extended range of mutations using next generation sequencing (NGS) has been proposed. Two approaches to NGS have been suggested: (1) a sensitive approach, which could result in fewer children with CF being missed following NBS, but more children with CRMS/CFSPID being identified; (2) a specific approach, which could result in more children with CF being missed following NBS, but fewer children with CRMS/CFSPID being identified [3]. Preliminary work undertaken by one newborn screening laboratory in England involved scoring CFTR variants of varying clinical consequence with a score of 1 per allele, and clearly pathogenic mutations with a score of 2 per allele. It was proposed that babies with CFTR gene mutations would therefore receive a combined score of 3 (if a sensitive approach were adopted) or 4 (if a specific approach were adopted) and would be referred for diagnostic follow-up. The purpose of this work was to gather, analyse, and compare the views of a range of stakeholders who have experience with CF about the proposed CF screening protocol incorporating NGS, particularly in relation to whether a sensitive or specific approach should be adopted. 

The research questions were as follows:How acceptable is the proposal to use expanded genomic testing in the newborn screening programme to people who have experience with CF/CRMS/CFSPID/carrier status?How are trade-offs made and valued between ensuring that we do not miss any cases of CF (a sensitive approach) and limiting uncertainty for some parents (a specific approach) and how is this viewed by different people and groups?How do people who have experience with CF/CRMS/CFSPID/carrier status view the avoidance of carrier identification and reporting within CF newborn screening using NGS (whichever score is used)?

## 2. Materials and Methods

### 2.1. Development of Data Collection Tools

The scope, structure, and data collection tools for the project were developed in collaboration with patient representatives, a CF clinical nurse specialist, a CF clinician, and representatives from the CF Trust, through involvement activities at the outset and at key stages of progress. Three parents of children with CF and one adult with a late CF diagnosis joined the Project Oversight Group. These parents met with the research team and other members of the oversight group prior to the start of the project to review and provide feedback on the proposed data collection approaches and techniques. Vignettes were constructed using interview data from a previous study [4] and were shared with members of our oversight group who provided feedback and suggestions that were incorporated into the final documents (please see Appendix A). 

### 2.2. Setting

The study sites included twelve CF centres in the UK.

### 2.3. Inclusion and Exclusion Criteria

To explore the acceptability of the inclusion and use of the proposed expanded genomic testing in the CF newborn screening programme, adults who had personal experience with CF, CRMS/CFSPID, or carrier status were included (group 1). Healthcare professionals (group 2) involved with CF and/or newborn screening were also invited to take part in the study. Staff who had not been involved with CF and/or newborn screening were excluded. Inclusion criteria can be seen in Table 1.

Recruitment and sampling: Group 1: adults who have personal experience with CF, CRMS/CFSPID, or carrier status. The data collection was driven by the principles of Grounded Theory [5,6]. Adults who had personal experience with CF, CRMS/CFSPID, or carrier status were purposefully sampled (i.e., based on a criteria defined prior to data collection) and theoretically sampled (i.e., based on emergent findings to ensure adequate representation of each participant group and in order to reach data saturation) [5,6]. Recruitment proceeded through various routes: CF centres via CF nurses and doctors, social media, and the CF Trust. CF nurses and doctors were able to provide potential participants with a study Participant Information Sheet (PIS) and give them the choice of either contacting the research team directly or consenting to share their contact details with the research team. 

Particular effort was made to ensure the inclusion of people whose first language is not English, Black, Asian, and minority ethnic (BAME) communities (to reflect the ethnic diversity of the CF population [7]), different socio-economic groups (to ensure inclusion of underserved populations), and men (as male perspectives are under-represented in newborn screening research—particularly fathers). 

Group 2: healthcare professionals involved with CF and/or newborn screening. Nationally, a two-stage sampling approach was employed that we have used successfully previously [8,9]. Professionals in health, education, and social care settings were first sampled purposively based on their experience with newborn screening and/or CF, followed by snowball sampling whereby participants were asked to suggest other eligible participants. As part of a wider programme of work, individuals were first invited to complete an anonymous survey about their views regarding the introduction of NGS in the CF newborn screening programme [10]. At the end of the survey, details were provided, enabling potential participants to contact the research team directly if they wished to be involved in the workshops. 

The project was regularly advertised via Twitter and groups recommended by the CF Trust were tagged. Individuals who appeared to have experience with CF were also searched on Twitter and tagged in Tweets. Professional societies (British Society of Genetic Medicine, British Thoracic Society, and Royal College of Midwives) were tagged in Tweets that included a link to take part in the project. International colleagues were recruited via existing collaborations with the European Cystic Fibrosis Society, CF Newborn Screening Special Interest Group, and collaborations with CF colleagues in America, Canada, and Australia. Written informed consent was obtained from all participants.

The format of the workshops was informed by members of the oversight group who felt that, due to the different experiences of the subgroups of Group 1 participants, it would be important to ensure they were not mixed to acknowledge any sensitivities around the different screening outcomes (i.e., CF vs. CRMS/CFSPID) and also to avoid associated potential anxiety (i.e., parents of children with CF discussing their child’s health with parents of children with CRMS/CFSPID). The oversight group felt that it would be appropriate to mix Group 2 participants (health professionals) among the Group 1 subgroups to allow collaborative discussion and reflect a more participatory approach between parents/adults and health professionals.

### 2.4. Data Collection

Online workshops were undertaken using Microsoft Teams (with an option to dial in). At the beginning of each workshop, a brief overview of NGS and the sensitive vs. specific approach was provided. In addition, as per the public dialogue [11], participants were informed that if the specific approach were to be adopted, and a child with CF were to be missed following NBS, they would likely be diagnosed clinically by the age of two years. The workshops were designed around the use of Q-methodology [12], which uses a statement-ranking technique (the Q-sort) to investigate the perspectives of participants. 

#### 2.4.1. Q-Set

As part of wider programme of work [13], prior online focus groups and interviews were conducted with adults and children who had experience with CF, CRMS/CFSPID, or carrier status, and an online survey was distributed to professionals involved with CF and/or newborn screening to ascertain their views about the anticipated impact of NGS on families and the National Health Service in the UK (NHS) [10]. A set of Q-statements reflecting dominant themes and illustrative quotes derived from the prior focus groups, interviews [13], and survey [10] were developed by the research team (the Q-set) and presented to participants prior to the workshop via Padlet (an online noticeboard).

#### 2.4.2. Q-Sorts

To explore how participants balanced the harms/benefits identified in the Q-set, and any variations in viewpoints, participants were asked, prior to the workshop, to individually rank (sort 1) statements from ‘extremely important’ (score 3) to ‘not important at all’ (score −3). During the workshop, participants came together to discuss their views and re-rank (sort 2) statements following their discussions.

#### 2.4.3. Willingness to Pay

The workshops also included a willingness-to-pay exercise [14] conducted via Poll Everywhere (an online polling tool); participants were asked to indicate the maximum amount they would be willing to pay for given scenarios related to the proposed CF screening protocol incorporating NGS.

### 2.5. Data Analysis

Q factor analysis was used to identify relationships between individual Q-sorts to illuminate differences and similarities in stakeholder perspectives [12]. Statements ranked as extremely important were given a score of 3, important 2, slightly important 1, neutral 0, slightly not important −1, not important −2, not important at all −3. Average pre-workshop and post-workshop scores for each of the statements were calculated. In addition, overall scores were calculated by adding up the average scores for each statement for each workshop to determine which statements were considered most and least important.

Data from the qualitative discussions were transcribed and a deductive approach to thematic analysis was used to detail and describe the patterns emerging from the data. Themes were generated using a latent approach [15].

## 3. Results

### 3.1. Q-Sort Statements

A set of Q-statements reflecting dominant themes derived from prior online focus groups, interviews [13], and an online survey [10], which were part of a wider programme of work, was developed (see Box 1) and presented to participants via Padlet.

Box 1Q-Statements.1. The uncertainty associated with CRMS/CFSPID can cause parents to
alter their life plans and restrict their child’s activities. 2.
Detecting more children with CRMS/CFSPID provides useful medical information.
3. A CRMS/CFSPID designation for a child can act as an entry way
in to the ‘CF world’ if needed. 4. Finding out children have CRMS/CFSPID,
provides useful health information for that child and the family. 5.
The uncertainty of CRMS/CFSPID is much less harmful than a missed CF
diagnosis. 6. Finding out your child has CRMS/CFSPID can act as a
safety net. 7. CRMS/CFSPID causes ongoing anxiety for parents. 8.
Having a CRMS/CFSPID designation may cause unnecessary anxiety for the child.
9. Identifying more children with CRMS/CFSPID would impact on NHS
resources. 10. Anxiety associated with CRMS/CFSPID decreases over
time. 11. If children with CF are missed during screening, it may
be harder for parents to get their concerns taken seriously later on. 12.
Identifying more children with CF/CRMS/CFSPID may mean we can reduce the
number of children with CF in the future. 13. It is vital that all
children with CF are picked up during screening so they can begin treatment
as early as possible’. 14. If more mutations are included in
screening, it may make it harder for health professionals to detect children
who may have been missed. 15. It is okay to identify more children
with CRMS/CFSPID if appropriate support, information, and communication is
available for those families. 16. The diagnostic process for CF
may be worse after a false negative screening result. 17. Missing
children with CF has a negative impact on parents and children. 18.
Missing children with CF may impact on child/parent relationships. 19.
Identifying more children with CRMS/CFSPID might improve knowledge and
awareness of CF and CRMS/CFSPID 20. CRMS/CFSPID is difficult to
explain to others, e.g., family, schools, health professionals.

A total of four workshops were conducted, with twenty-eight adults who had taken part in prior focus groups/interviews [13] and nine health professionals who had completed a survey [10] as part of a wider programme of work, as follows: two workshops with parents of children with CF (*n* = 9 and 12 for workshop 1 and 2, respectively), one workshop with parents of children with a CRMS/CFSPID designation (*n* = 3), and one workshop with adults with CF (*n* = 4). Two health professionals were present in each of workshops 1–3 and three health professionals were present in workshop 4. The workshops were audio recorded and transcribed. The composition and duration of the workshops and Padlet completion is summarised in Table 2.

Prior to the workshops, between 73 and 100% of participants completed the Padlets individually (sort 1). An overview of scores before (sort 1) and after (sort 2) the workshops for health professionals and parents of children with CF/CRMS/CFSPID and adults with CF can be seen in Table 3.

The greatest increases in ranking between individual (pre-workshop—sort 1) and agreed (post workshop—sort 2) scores were observed for ‘if children with CF are missed during screening, it may be harder for parents to get their concerns taken seriously later on’ (statement 11), ‘it is okay to identify more children with CRMS/CFSPID if appropriate support, information, and communication is available for those families’ (statement 15), and ‘the diagnostic process for CF may be worse after a false negative screening result’ (statement 16), indicating that participants viewed them as more important following the workshops. The greatest decreases in ranking between individual (pre-workshop) and agreed (post workshop) scores were observed for ‘uncertainty associated with CRMS/CFSPID can cause parents to alter their life plans and restrict their child’s activities’ (statement 1), ‘identifying more children with CRMS/CFSPID would impact on NHS resources’ (statement 9), ‘missing children with CF may impact on child/parent relationships’ (statement 18), and ‘CRMS/CFSPID is difficult to explain to others e.g., family, schools, health professionals’ (statement 20) indicating participants viewed them as less important following the workshops.

The highest scoring statements overall (most important) were: ‘the uncertainty of CRMS/CFSPID is much less harmful than a missed CF diagnosis’ (statement 5), ‘if children with CF are missed during screening, it may be harder for parents to get their concerns taken seriously later on’ (statement 11), ‘it is vital that all children with CF are picked up during screening so they can begin treatment as early as possible’ (statement 13), ‘it is okay to identify more children with CRMS/CFSPID if appropriate support, information and communication is available for those families’ (statement 15), and ‘missing children with CF has a negative impact on parents and children’ (statement 17). The lowest scoring statements overall (least important) were: ‘identifying more children with CRMS/CFSPID would impact on NHS resources’ (statement 9) and ‘CRMS/CFSPID is difficult to explain to others, e.g., family, schools, health professionals (statement 20).

### 3.2. Themes

#### 3.2.1. Information Provision and Communication

In terms of the sensitive approach to NGS, which involves identifying more children with a CRMS/CFSPID designation, it was felt that appropriate and timely information provision when communicating results would be key to alleviating undue anxiety. However, it was also felt that this needed to be accompanied by ongoing support to address any continuing uncertainty.


*“…with good education and with the right support and with lots of plenty of information for that family, all those families having that [CRMS/CFSPID designation], they could learn and adapt and it doesn’t have to be, it doesn’t have to be so life changing like it, it would obviously be anxiety causing but...with the right team of people explaining it and helping people manage it and how people would bring it into their life. Like it doesn’t actually have to be as big as that”.*
Parent of child with CF

In addition, it was felt that standardising information regarding a CRMS/CFSPID designation would also be important when communicating this screening outcome to parents if more children were potentially going to be identified with this outcome via the sensitive approach to NGS. It was recognised that for some health professionals, encountering a child with a CRMS/CFSPID designation could be quite unusual and therefore, providing standardised information could help those working in primary care, for instance, to manage parental expectations and concerns.


*“…finding some wording and description that can be used on a national basis. So, we’re doing things in a standardized way across the UK would be important…for a GP who may only ever have one or two CF patients in a working lifetime then CRMS/CFSPID is an even more nuanced thing for them to have to take on and deal with…we just need to have an agreed national framework for how we manage that and the language we use around it…”.*
Health professional

#### 3.2.2. Importance of Screening

The overriding principle that influenced parents’ decisions regarding whether the sensitive or specific approach to NGS for CF screening would be more appropriate was the importance of not missing children with CF due to the belief that it could be detrimental to the child’s health outcomes.

*“Although I have children with CFSPID, I still feel like the most important thing in all of this is to pick up the babies with CF”.*.Parent of child with CRMS/CFSPID

Similarly, when balancing the possible outcomes of the sensitive versus specific approach to NGS for CF screening, many participants considered the uncertainty associated with CRMS/CFSPID to be less harmful than a missed CF diagnosis.


*“The uncertainty with the CRMS/CFSPID, although it’s not nice and it’s hard to deal with as a parent, I think it’d be a hell of a lot harder to deal with a poorly child that’s got a missed CF diagnosis”.*
Parent of child with CRMS/CFSPID

Parents also favoured the sensitive approach to CF NGS as they valued the CRMS/CFSPID designation in terms their perception of it enabling them access to the CF team if needed.

*“To me the bigger benefit of the sensitive approach is for the patients who need the quick access and who struggle to get it when they need it…Having this big designation means you’re on a fast track to the right treatments. That, to me is much more important…”*.Parent of child with CF

The importance of increasing knowledge and understanding of CRMS/CFSPID by identifying more cases was also evident in the workshops.

*“…I also think the more evidence we’ve got about how the mutations present and how their presentations…if we do detect it for families that we give them the right support…detecting more children gives useful information”.*.Health professional

Although, there was also evidence of hesitancy in terms of adopting the sensitive approach to NGS due to the magnitude of the problems that could potentially be identified and the implications of this.

*“My concern is the whole process of going to next generation sequencing. We’re kind of opening a Pandora’s box. I feel that I’m not entirely sure we should, if I’m honest. I understand there are some benefits of it…[but] all the countries [in Europe] that had taken on next generation newborn screening were having real issues in terms of the just the numbers of patients with CFSPID”.*.Health professional

#### 3.2.3. Harms of Screening

##### Impact on Family due to the Diagnostic Odyssey

Identifying children with a CRMS/CFSPID designation was recognised to be anxiety-inducing due to the associated uncertainty, but most parents felt this would likely subside as the child got older and remained asymptomatic.

*“…we’ve only been living with this for three years, and… I feel like my anxiety is getting better”.*.Parent of child with CRMS/CFSPID

Most parents of children with a CRMS/CFSPID designation also described not knowing where they fit in terms of the CF world.

*“…when we was going through all the diagnosis, I like looked online for support groups obviously not wanting to, you know, stress out my clinic too much. Um, but I looked online for support groups and you, you’re kind of directed towards all the CF groups which [makes you feel] you’re like a fraud”.*.Parent of child with CRMS/CFSPID

The issue of a missed diagnosis of CF or not identifying a child with CRMS/CFSPID who later converted to a CF diagnosis or developed a CFTR-related disorder affecting parent–child relationships was also discussed. This was, to some extent, related to the guilt associated with not identifying that there was something wrong with the child and/or pushing harder for answers, but also to the time spent seeking answers and interrupting ‘normal’ family time.

*“…later in life, ‘well, why didn’t you push harder for tests? ‘You know that sort of reaction could come in. I’m thinking it’s not about so much how poorly they are, it’s more, why did the parent not do more to get the medical, the medical intervention and help”.*.Parent of child with CF

##### Identification of Carriers

Not identifying carriers if the CF screening protocol were to incorporate NGS was viewed as having both positive and negative consequences. Some participants viewed carrier status as important for parents’ reproductive decision making as well as information to share with family members to enable them to make choices in this regard.

*“…if you know about your genes then you have the option to decide what you do about it. And obviously, we wouldn’t take our children back because they are who they are now. However, if we’d known more, they’d still be who they are now. We just wouldn’t have known them as CFers…we just we wouldn’t have known them having CF because we’d have made that informed decision”.*.Parent of child with CF

Participants also described the guilt of unknowingly passing on a faulty gene to a child and therefore considered being aware of carrier status as helpful.

*“I got my carrier status from my mum, and she actually broke down in tears and cried because she obviously didn’t know she was a carrier, which meant obviously she feels partly to blame because obviously I’m a carrier from her and now my daughter has CF and she’s been in bits over it”.*.Parent of child with CF

However, others felt that carrier status was not really relevant to the aims of the CF screening programme, and some had found talking to family members about their CF carrier status difficult as they could be dismissive or not want to know.

*“…the point of doing the newborn screening is to pick up children that need medical intervention, not to pick up people that can then tell other people I’ve got CF and...I mean, it’s a sort of consequence of the screening, but it’s not the point of the screening”.*.Parent of child with CRMS/CFSPID

##### Impact on Clinical Team/Resource Implications

The potential burden of identifying more cases of CRMS/CFSPID, should the sensitive approach to NGS be adopted, on already stretched NHS resources was also discussed during the workshops. The importance and need to provide the required support to families following a CFSPID diagnosis to alleviate potential harm was emphasised by both parents and health professionals.

*“I totally understand that services are, you know at breaking point. But I sort of feel like…the support has to be there”.*.Parent of child with CRMS/CFSPID

*“…it’s difficult with stretched services, lack of psychology, lack of, you know, all of these things and maybe just the fact that people aren’t as aware of it and GPs aren’t aware of it and schools don’t know what it is, and nobody really understands it”.*.Health professional

In addition, most parents recognised the importance of offsetting the cost of identifying additional children with CRMS/CFSPID, should a sensitive approach to NGS be adopted, with missing more cases of CF, should a specific approach to NGS be adopted, and the additional tests and treatment that may be needed during the course of reaching a diagnosis, as well as the impact on the child’s health for not having been picked up in infancy.

*“…it’s worth arguing that the long-term strain on NHS, if things aren’t picked up, will have a really big impact. So, you know the interventions if children don’t have that early care, that preventative care and you know it might be a hospital admission. I mean would be very costly…”*.Parent of child with CF

There were mixed views from health professionals in terms of the impact on clinical teams. Some felt that picking up an additional 50–60 children per year (from 25–30 currently to 80 per year if the sensitive approach to NGS were to be implemented) equated to approximately 2–3 children per team and therefore may have only a minimal impact on service delivery.

*“…80 to one team would be a huge impact. Yeah, but one or two a year and we have runs like that”.*.Health professional

However, others considered the cumulative effect of identifying an additional 2–3 children per year and felt that this could become unmanageable over time if resources were to continue at current levels.

##### Staff Understanding and Communication with Families

Parents of children with CRMS/CFSPID discussed the difficulties of having their concerns taken seriously by health professionals outside of the CF team and therefore relying on the team to ensure their child received appropriate care and support in the community setting.

*“…when [Daughter’s name] was younger, she really struggled to fight off any cold or anything…so, you took her to the GP, they did not take you seriously. I feel like the only people that took us seriously was the CF team and then she got the help she needed. So, they wrote letters to the GP and said, look, she can’t fight it without the help. So that was the only way the GPs took me seriously”.*.Parent of child with CRMS/CFSPID

Most parents also thought that the adoption of the specific approach to NGS could make it even more difficult to get health professionals to listen to their concerns if their child were missed with CF/CRMS/CFSPID, as some health professionals might become overly confident in a negative diagnosis using NGS, which might make identifying a false-negative case harder.

*“…if CFSPID isn’t included, in some people’s minds, which says no, it’s not cystic fibrosis. So, I think it could make it more difficult for professionals to detect, whereas if you include the CFSPID it makes it, you know, it’s not off the radar in other, in other words, it’s still a consideration”.*.Parent of child with CF

In addition to standardising communication around the CRMS/CSPID designation, health professionals indicated that greater clarity around how to manage children with a CFSPID designation may be beneficial in terms of preventing clinicians from using CF protocols unnecessarily and therefore over-medicalising essentially healthy children as well as reducing disparities in the way children are managed nationally.

*“I worry probably a little bit more about when you have so much contact with the CF team…I think if you have a designation sometimes, I think people jump in very quickly with lots of tests, lots of investigations, lots of antibiotics which maybe there’s a possibility that’s not always needed…”*.Health professional

This was echoed in experiences of parents of children with CRMS/CFSPID.

*“I think it’s different in different places, and different teams will treat it differently. And even within the team, you know, I think I’ve always felt like some people take it extremely seriously and almost don’t differentiate it from CF for fear that you’re gonna miss something”.*.Parent of child with CRMS/CFSPID

### 3.3. Willingness-to-Pay Exercise

Thirty out of the thirty-seven participants who took part in the workshops completed the willingness-to-pay exercise: fifteen parents of children with CF, seven health professionals, four adults with CF, two parents who had received a false negative screening result for their child, and two parents of children with a CRMS/CFSPID designation.

Parents were asked six questions based on the Van Westerdorp Price sensitivity meter [14]. The first four questions and associated responses are shown in Table 4.

The fifth question asked participants to state the maximum amount they would be willing to pay for their baby to have extended genetic testing (next generation sequencing) for CF screening, if it were available privately. Responses ranged from, GBP 0 (*N* = 3 (10%)) to ‘as much as could be afforded’ (*N* = 2 (7%)) or GBP 25–30K (*N* = 1 (3%)). The final question asked whether participants preferred the sensitive or specific approach to NGS. Twenty-eight (93%) respondents stated they would prefer the sensitive approach. The two respondents (7%) who stated they would prefer the specific approach were both health professionals.

## 4. Discussion

Parents and professionals in this study favoured the sensitive approach to NGS if it were to be incorporated into the CF NBS protocol. However, as this had the potential to increase the number of cases of CRMS/CFSPID being identified, the importance of supporting parents adequately and appropriately after being informed of their child’s CFSPID designation was of paramount importance to alleviate any associated anxiety. Most parents also expressed a need for CRMS/CFSPID-specific support groups due to feeling as though they did not have a right to attend groups for parents with children with CF. This has been reiterated in the literature; parents of children with a CRMS/CFPISD designation have described confusion regarding whether they fitted into the ‘CF world’ or the ‘healthy kid world’ [4]. In addition, the workshops highlighted the importance of standardising information so that all families received the same messages. The importance of standardising information about newborn screening results has been highlighted in previous research [16]. Health professionals also thought it would be important to ensure that those managing children with a CRMS/CFSPID designation had adequate knowledge and understanding to enable them to do this effectively. However, it was acknowledged that currently there is not enough data to inform management [2,17,18] and that for some health professionals (e.g., GPs), they may encounter CRMS/CFSPID so rarely that it is difficult to stay informed.

The most important influencing factor in terms of decision making between the sensitive and specific approach to NGS for CF screening was ensuring children with CF are not missed (Q-sort statements 11,13,14,16). While participants appreciated that CRMS/CFSPID could be anxiety provoking, parents and most health professionals felt it was less damaging to detect a child with a CRMS/CFSPID designation than to miss a child with CF. As per the public dialogue [11], it was explained to parents that if the specific approach were to be adopted, this would result in more children with CF being missed. Some professionals indicated that a delayed diagnosis of CF until two years of age would not result in significant long-term harm. However, most parents felt this would not be acceptable due to their lived experiences of their child becoming symptomatic before this age, and their experience of the benefit of protein modulator therapies. Additionally, parents of children who had experience with CF, adults with CF, and health professionals who completed the online survey discussed the psychological damage associated with a delayed diagnosis. Furthermore, health professionals who took part in the workshops did not agree with the view that a delayed diagnosis of CF until two years of age would not result in significant long-term harm. This has been supported by previous studies, which have demonstrated that lung function may already be abnormal in 3-month-old infants with CF diagnosed by newborn screening [19] and that using treatment protocols are important during early life [20]. This supports the need for the early identification and management of children with CF to ensure the best long-term health outcomes.

Parents also expressed concern about the diagnostic odyssey that may occur if a child with CF was missed at screening, or if a child with CRMS/CFSPID who had not been detected following screening later became symptomatic. There are few studies that have explored the impact of a late diagnosis both clinically and psychologically. However, the limited evidence that is available [21,22] in conjunction with anecdotal reports suggest that this can result in increased anxiety, guilt, anger, and mistrust in health services. These are important considerations when deciding between the sensitive and specific approach to NGS for the CF screening protocol and the associated outcomes in terms of increasing or decreasing the number of children identified with a CRMS/CFSPID designation and the number of children with CF who may be missed.

Parents and health professionals in the present project also indicated that picking up more children with CRMS/CFSPID could be advantageous in terms of increasing knowledge about different genotypes and how best to treat them over time. Evidence for managing children with CRMS/CFSPID is still evolving; those who convert to a CF diagnosis may benefit from early intervention to prevent long-term complications [2]. But it is unclear which children this applies to or how frequently they should be monitored [2,17,18]. Parents and health professionals in the current project felt that identifying more children with a CRMS/CFSPID designation could be advantageous for increasing knowledge of this designation in terms of the different phenotypes and their clinical significance as well as subsequent management. Views regarding the significance of specific mutations are divided. While some postulate that this could be helpful for predicting the risk of disease evolution [23], others suggest that due to the spectrum of clinical heterogeneity, this may be less useful [24]. In many studies, the confirmation of conversion to a CF diagnosis was often due to a positive sweat chloride result (i.e., ≥60 mmol/l) either as part of the review process or because the child had become symptomatic [17,25,26,27,28].

The current CF screening protocol was designed to minimise the number of CF carriers unavoidably detected [1]. The introduction of NGS into the CF screening protocol using either the sensitive or specific approach would mean that carriers would be no longer reported. There was some concern expressed by adults who had experience with CF in the present project regarding the potential implications of carrier status and the CRMS/CFSPID designation. As only a proportion of the CFTR mutations would be included in the proposed plans for NGS, some were worried that a ‘probable carrier’ status had the potential to be a possible CRMS/CFSPID designation. This was also considered by health professionals who completed the online survey who stated that ‘probable carrier’ status may not actually be a benign state.

Strengths and Limitations: Although the number of health professionals involved (*n* = 9) was small, they represented six geographical areas in the UK, which increases the transferability of the findings. Health professionals self-selected whether they wanted to be involved in the workshops; those with a pre-existing interest in this topic may have been more likely to self-select into the study. These health professionals may have different experiences in relation to managing children with a CRMS/CFSPID designation or a missed CF case compared to those who did not participate in the study. The study design, data collection, and analysis were influenced by members of the patient and public involvement oversight group. This included parents of children with CF, an adult with a late CF diagnosis, a CF clinical nurse specialist, a CF clinician, as well as a representative from the CF Trust.

Recommendations for practice: The findings from this study suggest a number of recommendations for practice. For example, at the time of NBS for CF, parents should be advised that, as well as identifying whether the child has CF or not, NBS for CF also has the potential to identify children with a CRMS/CFSPID designation; parents should be made aware that the outcome may not be binary. Specific support strategies for parents with children with a CRMS/CFSPID designation should be developed. While parents are advised that their child with CRMS/CFSPID does not have CF and should be viewed as ‘healthy’, the uncertainty associated with this outcome can lead to increased anxiety. In addition, parents do not necessarily feel that they ‘fit’ in the ‘CF world’ and can be unsure where to seek support. A health economic analysis of the actual costs associated with managing a CRMS/CFSPID designation as well as identifying a missed case of CF or CRMS/CFSPID should be conducted. Research should be conducted to inform definitive written guidelines for health professionals regarding the clinical management of children with a CRMS/CFSPID designation.

## 5. Conclusions

Parents and health professionals expressed a preference for a sensitive approach to NGS if it were to be incorporated into the CF screening protocol. Given that this would result in more cases of CRMS/CFSPID being identified, the possibility of being given a CRMS/CFSPID designation (rather than CF, carrier, or unaffected) needs to be adequately explained at the time of screening, and specific support should be available for families following screening. While identifying more children with a CRMS/CFSPID designation may have resource implications, this could be offset by reducing the number of families who experience the diagnostic odyssey often associated with a missed diagnosis of CF or who have a child who has not been identified with CRMS/CFSPID but who may develop symptoms later in life. More research is needed to inform definitive guidelines for the management of children with a CRMS/CFSPID designation.

## Figures and Tables

**Table 1 IJNS-10-00013-t001:** Participants and inclusion criteria.

Type of Participant	Sub-Category
GROUP 1	
Adults aged over 18 years who met the following criteria:	(i) Were diagnosed with CF in childhood (ii) Were diagnosed with CF as adults (including those who would have been identified as CRMS/CFSPID if newborn screening had been available and this outcome was reported (iii) Are carriers of CF (including parents/relatives of children/adults with CF and adults identified via other routes, e.g., private testing)
Parents (aged over 18 years) of children identified through newborn screening who met the following criteria:	(iv) Have CF (v) Are carriers of CF (vi) Have a CRMS/CFSPID designation (vii) Have received a false negative newborn screening result for CF
GROUP 2	
Health Care Professionals who met the following criteria:	Involved in processing, communicating positive newborn screening results for CF to families, or supporting families in health, community, or education settings.

**Table 2 IJNS-10-00013-t002:** Composition and duration of Workshops.

Workshop	Number of Non-Health Professionals	Number of Health Professionals	Duration Median (Range)	% of Padlets Completed Prior to Workshop
1	Nine parents of children with CF	Two CF doctors	92.98 min (88.15–97.8 min)	73
2	Twelve parents of children with CF	One screening nurse, one genetic counsellor		71
3	Three parents of children with a CRMS/CFSPID designation	One CF doctor, one CF CNS	107.12 min	100
4	Four adults with CF	Three CF doctors	110.13 min	86

**Table 3 IJNS-10-00013-t003:** Summary of Scored Q-Sort Statements.

	Workshop 1			Workshop 2			Workshop 3			Workshop 4			Overall Average			Overall Score
	Sort 1		Sort 2	Sort 1		Sort 2	Sort 1		Sort 2	Sort 1		Sort 2				
	CF Parents	HPs	Agreed Score	CF Parents	HPs	Agreed Score	CRMS/CFSPID Parents	HPs	Agreed Score	CF Adults	HPs	Agreed Score	Parents/Adults	HPs	All	
1. CRMS/CFSPID uncertainty causes parents to alter life plans/restrict child’s activities	1	2	1	1	1	−2	2	3	2	1	2	1	1	2	1	14
2. Detecting more children with CRMS/CFSPID provides useful medical information	1	1	2	3	2	2	2	2	1	3	0	0	2	1	2	17
3. A CRMS/CFSPID designation acts as an entry way in to the ‘CF world’	1	2	2	2	2	2	1	1	0	2	1	3	2	1	2	19
4. A CRMS/CFSPID designation provides useful health information for child and family	2	2	2	3	2	3	2	1	0	3	−1	0	2	1	2	18
5. Uncertainty of CRMS/CFSPID is less harmful than a missed CF diagnosis	2	3	3	3	2	3	2	3	3	3	1	1	2	2	2	27
6. The CRMS/CFSPID designation acts as a safety net for parents	1	1	2	3	1	2	0	3	1	1	1	3	1	1	2	18
7. CRMS/CFSPID causes ongoing anxiety for parents	1	2	1	1	2	−2	2	3	1	1	3	3	1	2	1	16
8. A CRMS/CFSPID designation causes unnecessary anxiety for the child	−1	2	1	0	−1	−2	1	2	1	0	2	1	0	1	0	7
9. Detecting more children with CRMS/CFSPID impacts NHS resources	−2	−1	−3	0	−1	0	−1	0	−2	1	−1	0	0	−1	0	−10
10. Anxiety associated with CRMS/CFSPID decreases over time	1	*	1	2	0	1	−1	1	0	1	2	0	1	1	1	7
11. A missed CF diagnosis makes it harder for parents to get taken seriously later in their child’s journey	1	3	3	2	2	3	2	2	3	3	2	2	2	2	2	27
12. Detecting more children with CF/CRMS/CFSPID may reduce prevalence of CF in future	1	0	1	2	1	3	0	0	−2	2	−1	1	1	0	1	7
13. Detecting children with CF during screening means they can begin treatment quickly	3	3	3	3	3	3	2	3	3	3	3	3	3	3	3	35
14. Including more mutations in screening makes it harder for HPs to detect missed diagnoses	1	3	−1	−1	1	2	0	1	0	1	1	2	0	1	0	9
15. A CRMS/CFSPID designation is okay if families get appropriate support and information	2	1	3	3	1	2	2	3	3	2	2	3	2	2	2	27
16. The diagnostic process for CF is worse after a false negative screening result	1	0	3	2	2	2	1	2	3	2	3	2	1	2	1	21
17. Missing children with CF has a negative impact on parents and children	3	3	3	2	3	3	2	3	3	2	2	3	2	3	2	31
18. Missing children with CF impacts child/parent relationships	3	3	−1	1	1	2	0	3	2	2	1	0	1	2	1	15
19. Identifying more children with CRMS/CFSPID improves knowledge/awareness of CF/CRMS/CFSPID	2	−1	2	2	0	3	2	2	2	2	2	1	2	1	2	18
20. CRMS/CFSPID is difficult to explain to others, e.g., family, schools, health professionals	−1	0	−2	1	−1	−3	0	2	−3	1	1	2	0	0	0	−4

Abbreviations: ‘Sort 1′ = average scores pre-workshop; ‘Sort 2′ = agreed scores post-workshop; ‘HP’ = health professional; * = not applicable.

**Table 4 IJNS-10-00013-t004:** Response to willingness-to-pay exercise.

Per Baby						
£1–50	£51–100	£101–200	£201–300	£301–400	£401–500	£500+
1. If paying privately, what price would you consider extended genetic testing (next generation sequencing) for CF screening to be so expensive that it is not a feasible option? (*n* = 30)						
*N* = 0 (0%)	*N* = 2 (7%)	*N* = 6 (20%)	*N* = 5 (17%)	*N* = 4 (13%)	*N* = 1 (3%)	*N* = 12 (40%)
2. If paying privately, what price would you consider extended genetic testing (next generation sequencing) for CF screening to be priced so low that you would feel the quality couldn’t be very good? (*n* = 28)						
*N* = 10 (36%)	*N* = 8 (28%)	*N* = 6 (21%)	*N* = 3 (11%)	*N* = 1 (4%)	*N* = 0 (0%)	*N* = 0 (0%)
3. If paying privately, what price would you consider extended genetic testing (next generation sequencing) for CF screening is starting to get expensive, so that it is not out of the question, but you would have to give some thought to using it? (*n* = 30)						
*N* = 1 (3%)	*N* = 2 (7%)	*N* = 3 (10%)	*N* = 3 (10%)	*N* = 13 (43%)	*N* = 5 (17%)	*N* = 3 (10%)
4. If paying privately for extended genetic testing (next generation sequencing) for CF screening, what is the highest price you would pay and still consider it to be good value for money? (*n* = 30)						
*N* = 0 (0%)	*N* = 2 (7%)	*N* = 7 (23%)	*N* = 7 (23%)	*N* = 3 (10%)	*N* = 6 (20%)	*N* = 5 (17%)

## Data Availability

The data presented in this study are available on request from the corresponding author. The data are not publicly available due to ethical constraints.

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
