# Peer review of "Stakeholder Views of the Proposed Introduction of Next Generation Sequencing into the Cystic Fibrosis Screening Protocol in England"

_2409-515X, 2024, doi:10.3390/ijns10010013_

Round 1

Reviewer 1 Report

Comments and Suggestions for Authors

IJNS gCF England (Chudleigh as senior author Comments  v1 01.06.24 

Thank you to the authors for undertaking this interesting and important study, one that is both nuanced to the condition (CF) and has some broader applicability. This reviewer supports publication in the journal once the below comments are addressed. Most are relatively minor, yet a few denoted by stars reflect more major-level revisions. I am happy to re-review should the authors and editors request.

NOTE: This reviewer did NOT undertake an Analysis or Verification of the data in the tables or  text

Abstract: 

Lines 14-16: from this reviewer’s perspective, the first sentence would benefit from modifying the placement of these clauses. As well, please specify the location of the proposed protocol (while notes location of participants in later sentence, this should not be assumed to be policy proposal location).

Lines 19-20: to ease readability of participant groups and numbers, suggest placing N’s in brackets within the larger parens rather than currently having N’s in parens within large paren.

Lines 19-20:  readers unfamiliar with CF may find it confusing when the abstract says “an uncertain outcome following screening for CF (termed CFSPID) “ yet there is no letter U to suggest Uncertainty. Perhaps rephrase to say as elsewhere “CF Screen Positive – Inconclusive Diagnosis  (CFSPID) “ and then add to beginning or end: /uncertain outcome

Lines  48-53: to ease readability, please consider another format without so much text in parens. Perhaps bullets or numbering eliminating most parens:  “….re-classified as ‘CF not suspected’ (i.e. a false positive screening test result, the baby is then discharged); CF confirmed (appropriate on-going treatment is then arranged); CF screen positive – inconclusive diagnosis (CFSPID) (where the screening result is positive, but the confirmatory tests are equivocal and there is insufficient information to classify the baby as having CF but neither is there sufficient clarity to discharge the baby. In these situations, some on-going monitoring is usually arranged) 

Lines 63-64: Some readers may need a refresher or are relatively unfamiliar with terms sensitive approach and specific approach. Please define them operationally before writing “score of 3 (if a 64 sensitive approach were adopted) or 4 (if a specific approach were adopted)” 

--they are also not clearly defined in Research Question 2

Line 88: “Vignettes constructed”. Please include the vignettes in the manuscript and indicate where can be found (please see XXXX)

Lines 103-4: Please clarify terms for unfamiliar readers: “purposefully and theoretically sampled” – most likely unfamiliar with ”theoretical” sampling; and not everyone familiar with “purposeful”

Line 110: first mention of  ‘CF world’ (excluding abstract)—suggest term ‘CF world’ introduced earlier with mention of participants and/or inclusion criteria

Line 113: Please spell out BAME as not all readers may be familiar

Line 143: Please clarify Q as not all readers will be familiar Q-methodology and Q-set (preferably in method section)

**Table 3 is challenging to read, especially without a header indicating what the statement said . From this reviewer’s perspective, the number doesn’t suffice unless a reader goes back and forth to Figure 1. Will be too much effort for some/many and then not read or understood. Please consider breaking up into sections with brief header summarizing each statement

Lines 206-229: Similarly would benefit from having a revised Table 3. It is currently overly challenging to read. Additionally much is in parens and gets lost. Suggest flipping here so that that statement Numbers be in parens with the actual statement outside of parens. From this reviewer’s perspective, the statement is more meaningful to the reader than the number.  

Line 259: Please delete comma

Line 270: A word seems to be missing (“Parents also favoured the sensitive approach the CF NGS as they valued the….”)

Line 282: “Although” appears dangling to me; please either add a comma, or a word to complete the sentence, or delete this word

**The authors often reference statements with “parents” and, in some contexts, this resonates as if they were all the parents. Please confirm that you intend this to reflect 100% of the parents, or else, please use a qualifier to descriptively modify quantities, frequencies, and subgroups (i.e., most, more than half, often, typically, parents of children with XYZ, etc)

Two examples, please see

Line 293: “…but parents felt this would likely…” Also line 433 “parents and most health professionals felt 

**From this reviewer’s perspective, the section “Harms of Screening” is far too lengthy a run (100 lines) without subsections that reflect analytic differentiation or commonalities.  Important points get lost as they are blended into each other. Please break this section -- Harms --into subthemes.

Table 4 Response to Willingness to Pay Exercise: From this reviewers’ perspective, readability would benefit from some reformatting to have a break between a response and the next question.

Line 406-8: Please clarify whether the fifth and last questions (in text body; outside of Table 4) are the precise questions asked or rephrased. If precise, then please add quotes; if not . please remove the comma after “asked” : “The fifth question asked, if it were available privately, what is the maximum amount you would be willing to pay for your baby to have extended genetic testing (next generation sequencing) for CF screening? 

** Please note a bit more of the “ CF world“ distinction  (nothing between 110 and 420) of parents of children with a CFPISD designation have described confusion regarding whether they fitted into the ‘CF world’ or the ‘healthy kid world’ from Line 421-2

Many sentences are overly complex and would benefit from dividing. For example:

*Lines 454-460: This single sentence is six lines long and overly complex. Would benefit from constructing into multiple smaller sentences. 

Lines 502-504: The last clause is confusing to this reviewer, and does not seem to align with the other clauses, particularly with ”but” – “While parents are advised their child with CFSPID does not have CF and should be viewed as ‘healthy’, the uncertainty associated with this outcome can lead to increased anxiety, but parents do not necessarily feel they ‘fit’ in the ‘CF World’. 

Line 514: “…outcome needs to be adequately explained at the time of screening…” 

Citations 9, 10, others: Please confirm that all of the citations referenced by a author of this paper is necessary.

Thank you

Author Response

We would like to thank the reviewer for the positive feedback and helpful comments that supported the revised manuscript. We have made every attempt to fully address these comments and believe the manuscript has benefited as a result. Details on how each of the reviewer's comments were addressed can be found on the attached 'Response to Reviewers' document.

Reviewer 2 Report

Comments and Suggestions for Authors

This manuscript by Holder et al. discusses the application of next-generation sequencing (NGS) for newborn screening of cystic fibrosis (CF). Parents of CF-positive or -carrier children, adult CF patients, and health care providers involved with CF and/or newborn screening, were the three stakeholder groups surveyed. Opinions from the different groups were gathered, analyzed, and compared through a systematic Q-method. Firstly, the author group, consisting of CF patient representatives, specialists, and experts, composed a concourse of major opinions on NGS application to CF screening from interviews with adult CF parents and patients and online surveys of healthcare providers involved with CF and/or newborn screening. Secondly, participants who showed interests in further participation from the interviews and online surveys were asked to sort a list of views on CF screening with NGS by whether they agreed, were neutral towards, or disagreed with the presented views. The sorting process was performed in two phases: the first one independently, and the second one among a group discussion. Finally, the authors scored the different views on CF screening with NGS to identify major opinions and themes. This manuscript also gave a comprehensive background on the advantages and disadvantages of current and proposed NGS-based CF screening methods, as well as detailed analyses and future suggestions based on the major views and themes reported.

The manuscript can be improved in the following ways: 

1.     State in the Materials and Methods section that this study will follow a Q-method and give a brief description of the method. The authors should consider structuring the Materials and Methods section in a standard Q-method format (concourse; q-set; p-set – includes settings, inclusion/exclusion & sampling groups; q-sorts; and data analysis).

2.     The ranking method used in this study needs to be elaborated upon in the Materials and Methods section for readers to understand the meanings of the Q-sort scores in Table 3 of Results.

3.     The authors should explain the calculation method and discuss the meaning of the Overall Score in Table 3 of Results because this measurement was a major determinant of which views were deemed most important to the participants from this study.

4.     Statement 14 in the q-sort statements said: “If more mutations are included in screening, it may make it harder for health professionals to detect children who may have been missed.” This statement essentially boiled down to the concern that healthcare providers might become overly confident in a negative diagnosis by NGS, which might make identifying a false-negative case harder. The main point of Statement 14 was not clear upon initial reading in the Results section. The context/discussion for this view came much later in the discussion section and without direct link to this statement and its scores from the Results section.

5.     The authors mixed participants from group 1 and group 2 in each of their four workshops to generate a consensus ranking of the Q-sort statements. However, group 1 consisted of distinct subgroups: CF parents, CFSPID (CF screen positive – inconclusive diagnosis) parents, and adult CF patients, but the authors conducted separate workshops with different subgroup of participants in group 1 (alongside participants in group 2 – healthcare providers) instead of mixing these subgroups of group 1 together in each workshop. The authors should explain clearly in the Materials and Methods section the reasoning for this separation of subgroups in group 1 participants.

6.     The authors should also state clearer in the introduction why they decided to mix group 2 and different subgroups of group 1 together for the workshops to generate consensus ranking. Also, they should discuss the advantage of this group assignment in the beginning of the Discussion section.

7.     In the second paragraph of Discussion section, the authors stated that “As per the public dialogue [17], it was explained to parents that if the specific approach were to be adopted, resulting in more children with CF being missed, views of some professionals indicated that a delayed diagnosis of CF until two years of age would not result in significant long-term harm.” It should be made clear if this information was explained to the participants during the workshops to generate consensus rankings as part of a planned set of information given to all participants during all workshops, or it was added spontaneously. If it was delivered during the workshops as part of the agenda, the authors must add this detail, and all other information provided to the participants by the researchers, into the Materials and Methods section. Information provided by researchers during the ranking workshops might have impacted participants’ decisions. If the piece of information was added spontaneously, the workshops and timeframes that the information was given must be stated clearly.

Author Response

(The authors gave the same response as above.)
